# The Closed-Loop Control of the Half-Bridge-Based MMC Drive with Variable DC-Link Voltage

Mauricio Espinoza [1,*], Matias Diaz [2], Enrique Espina [2], Andrés Mora [3], Arturo Letelier [4], Felipe Donoso [4] and Roberto Cárdenas [4]

1 School of Electrical Engineering, Universidad de Costa Rica, San José 11501-2060, Costa Rica
2 Department of Electrical Engineering, Universidad de Santiago de Chile, Avenida Ecuador 3519, Santiago 9170124, Chile; matias.diazd@usach.cl (M.D.); enrique.espinag@usach.cl (E.E.)
3 Department of Electrical Engineering, Universidad Técnica Federico Santa María, Av. España 1680, Valparaiso 234000, Chile; andres.mora@usm.cl
4 Department of Electrical Engineering, Universidad de Chile, Santiago 8370451, Chile; aletelier@ing.uchile.cl (A.L.); fdonoso@ing.uchile.cl (F.D.); rcardenas@ing.uchile.cl (R.C.)
* Correspondence: mauricio.espinoza_bola@ucr.ac.cr; Tel.: +506-2511-2639

**Abstract:** The modular multilevel converter (MMC) based on half-bridge modules is a power converter topology suitable for high-power medium-voltage variable-speed drives. However, the voltage of its flying capacitors is negatively affected when low frequencies appear at the AC port. This paper analyzes the influence of using a variable DC port voltage in a machine-side MMC by implementing a closed-loop approach, ensuring a constant voltage fluctuation in the capacitors of the MMC during the whole operating range. The effectiveness of the proposed control scheme is demonstrated through simulation studies and experimental validation tests conducted using a 7.5 kW experimental prototype composed of an induction machine fed by an MMC with 18 half-bridge cells.

**Keywords:** modular multilevel converter; machine drive; dynamic modeling





## 1. Introduction

In the early period after the invention of the modular multilevel converter, research efforts were mainly focused on high-voltage direct current (HVDC) applications of this topology. However, considering all the advantages of the MMC, such as full modularity, low total harmonic distortion (THD) in the currents, and the simplicity of achieving redundancy, MMCs have been recently proposed for the control of electrical machines in applications such as railway traction and drive systems [1–6]. For drive applications, the MMC is considered suitable for quadratic-torque speed profile loads, where better performance has been reported [7,8].

Figure 1 depicts an MMC-based drive. Figure 1a shows the MMC and six clusters. Each cluster is composed of *n* half-bridge modules (see Figure 1b) and an inductance device. Because of the flying capacitors, one of the most important tasks of the control systems is to maintain the voltage in each capacitor operating within an acceptable range. This control target is difficult to fulfill when the machine is operating at low electrical frequencies because the capacitor voltage variations are unstable [3,4]. Therefore, for control purposes, the operating range of the MMC is usually divided into the high-frequency mode (HFM) and the low-frequency mode (LFM).

During the LFM, the common-mode voltage and the circulating currents have been proposed to completely eliminate the voltage fluctuations of the MMC capacitors [3–6]. Another methodology to reduce the magnitude of the mitigation signals is to use "margin-based" control strategies. In this case, the low-frequency voltage oscillations are not regulated to zero, because this requires large circulating currents as the machine frequency

increases [2,9,10]. Instead, the voltage fluctuations are allowed to operate within an acceptable margin around the DC voltage of the capacitor.

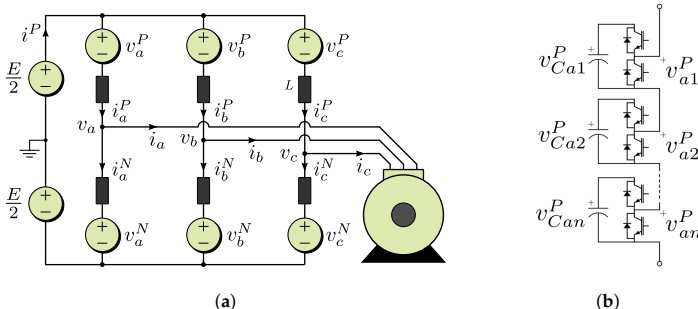

<div align="center">(a)          (b)</div>

**Figure 1.** MMC-based drive. (**a**) Converter topology; (**b**) cluster.

Recently, the DC port voltage $E$ (see Figure 1) has been manipulated to maintain the amplitude of the voltage fluctuations in the capacitors [11–15]. In this case, the grid-side converter is an active rectifier [11,12,16,17] or an MMC composed of half-bridge and full-bridge cells [18,19]. However, most of the proposals add full-bridge modules to the machine-side MMC [14,15] or reduce $E$ to values close to zero during machine start-up, limiting the dynamic response of the converter [11–13]. Additionally, in all these studies, the DC port voltage is defined using feed-forward signals. Consequently, the dynamic relation between the capacitor voltages and the DC port voltage $E$ is not considered.

As expected, these control systems produce different dynamics in the converter capacitors and in the cluster voltage (sum of the capacitor voltages in the same cluster). As a representative example, Figure 2 shows the behavior of the oscillating component (without DC value) of the cluster voltage $\tilde{v}_{Cx}^{X}$ (with $X = P$ or $N$ and $x = a$, $b$ or $c$, see Figure 1). Additionally, the amplitude of $\tilde{v}_{Cx}^{X}$, $|\tilde{v}_{Cx}^{X}|$, is also depicted in Figure 2, where the gray area represents the LFM, and the same scale in the y-axis has been used. Based on these definitions, the control strategies discussed above can be classified as follows.

1. **Full mitigation**: the low-frequency fluctuations in the cluster voltages are regulated to zero during LFM, producing large circulating currents [3–6]. See Figure 2a.
2. **Margin-based**: a voltage fluctuation is allowed in the cluster voltages during LFM. The peak value of the circulating currents is reduced but a large common-mode voltage is still required [2,9,10]. See Figure 2b.
3. **Variable DC port voltage**: the DC port voltage $E$ is used to maintain the amplitude of the fluctuations in the cluster voltages. However, the machine-side MMC has more components [14,15] and its dynamic regulation is poor [11–13]. See Figure 2c.

In this work, these issues are discussed and the MMC response for a varying DC port voltage is studied, providing the following contributions to the state of the art.

1. A small-signal model is developed to fully analyze the dynamics of the MMC when the magnitude of the DC port voltage $E$ is modified.
2. A closed-loop control system is proposed for MMC-based drives. Using this scheme, the DC port voltage is modified considering the dynamics of the capacitor voltages and not only steady-state expressions.
3. The proposed approach reduces the common-mode voltage during LFM, reducing the damage to the machine insulation or the leakage currents in the bearing [20,21].

Experimental results obtained with a 7.5 kW prototype are described to validate the effectiveness and feasibility of the proposed control strategies. The experimental prototype is composed of an induction machine fed by an 18-cell MMC and an active rectifier composed of 12 half-bridge and 6 full-bridge cells. The experiments include steady-state as well as dynamic performance tests, considering (a) variations in the DC link voltage when the machine is used from 0 to nominal speed and (b) comparisons between the system variables when a maximum or minimum DC port voltage is utilized. The reduction in the

common-mode voltage is also experimentally demonstrated when the proposed control strategy is implemented.

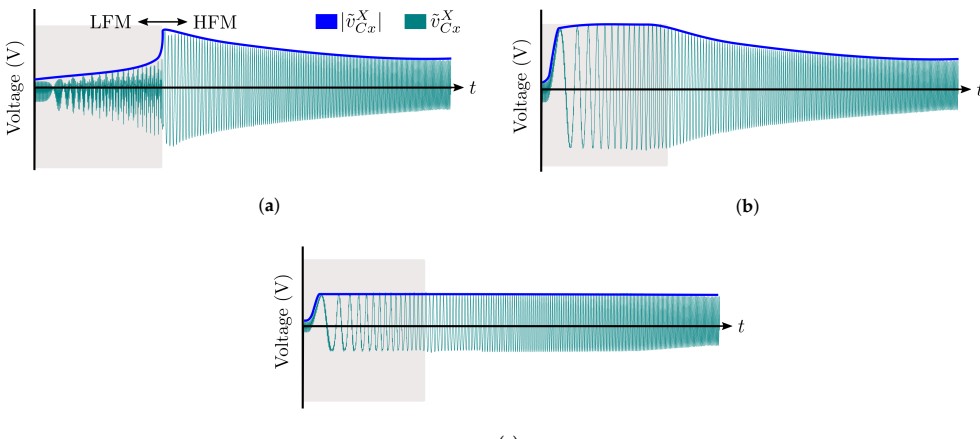

**Figure 2.** Evolution of the MMC-based drive control strategies. (**a**) Full mitigation; (**b**) margin-based mitigation; (**c**) variable DC port voltage.

## 2. Conventional Modeling of the MMC

In this work, the MMC modeling presented in [2,5,22] is used. For completeness, it is briefly discussed in this section. Representing the output cluster voltages as voltage sources (see Figure 1), the dynamics of the MMC cluster currents are obtained, applying the Kirchhoff voltage law as follows:

$$
L\frac{d}{dt}\overbrace{\begin{bmatrix} i_a^P & i_b^P & i_c^P \\ i_a^N & i_b^N & i_c^N \end{bmatrix}}^{:=\mathbf{I}_{abc}^{PN}} = -\overbrace{\begin{bmatrix} v_a^P & v_b^P & v_c^P \\ v_a^N & v_b^N & v_c^N \end{bmatrix}}^{:=\mathbf{V}_{abc}^{PN}} + \frac{E}{2}\begin{bmatrix} 1 & 1 & 1 \\ 1 & 1 & 1 \end{bmatrix} + \begin{bmatrix} -v_a & -v_b & -v_c \\ v_a & v_b & v_c \end{bmatrix} \tag{1}
$$

where the $\Sigma\Delta\alpha\beta 0$-transformation can be applied to (1) to allow the decoupled control of the converter currents [5,6]. This transformation modifies a $2 \times 3$ matrix from the *PNabc*-coordinated space to the $\Sigma\Delta\alpha\beta 0$ one through the following matrix transformation:

$$
\mathbf{X}_{\alpha\beta 0}^{\Sigma\Delta} = \mathbf{C}^{\Sigma\Delta} \cdot \mathbf{X}_{abc}^{PN} \cdot \mathbf{C}_{\alpha\beta}^{\mathsf{T}} \tag{2}
$$

where

$$
\mathbf{C}^{\Sigma\Delta} = \begin{bmatrix} \frac{1}{2} & \frac{-1}{2} \\ 1 & -1 \end{bmatrix} \quad \text{and} \quad \mathbf{C}_{\alpha\beta}^{\mathsf{T}} = \begin{bmatrix} \frac{2}{3} & \frac{-1}{3} & \frac{-1}{3} \\ 0 & \frac{1}{\sqrt{3}} & \frac{-1}{\sqrt{3}} \\ \frac{1}{3} & \frac{1}{3} & \frac{1}{3} \end{bmatrix}^{\mathsf{T}} \tag{3}
$$

Applying this transformation, (1) results in

$$
L\frac{d}{dt}\overbrace{\begin{bmatrix} i_\alpha^\Sigma & i_\beta^\Sigma & \frac{1}{3}i^P \\ i_\alpha & i_\beta & 0 \end{bmatrix}}^{:=\mathbf{I}_{\alpha\beta 0}^{\Sigma\Delta}} = -\overbrace{\begin{bmatrix} v_\alpha^\Sigma & v_\beta^\Sigma & v_0^\Sigma \\ v_\alpha^\Delta & v_\beta^\Delta & v_0^\Delta \end{bmatrix}}^{:=\mathbf{V}_{\alpha\beta 0}^{\Sigma\Delta}} - 2\begin{bmatrix} 0 & 0 & -\frac{1}{4}E \\ v_\alpha & v_\beta & v_0 \end{bmatrix} \tag{4}
$$

In this coordinated system, the machine currents ($i_\alpha$ and $i_\beta$) and voltages ($v_\alpha$, $v_\beta$ and $v_0$) are expressed in $\alpha\beta 0$-coordinates; $i_\alpha^\Sigma$ and $i_\beta^\Sigma$ are the circulating currents and $i^P$ is the DC port current. Notice that each current in $\mathbf{I}_{\alpha\beta 0}^{\Sigma\Delta}$ can be regulated by manipulating a voltage in the matrix $\mathbf{V}_{\alpha\beta 0}^{\Sigma\Delta}$.

In addition to the previous dynamic model, the relationship between the cluster instantaneous power and the capacitor voltage is given by

$$C\bar{v}_C \frac{d}{dt} \underbrace{\begin{bmatrix} v_{Ca}^P & v_{Cb}^P & v_{Cc}^P \\ v_{Ca}^N & v_{Cb}^N & v_{Cc}^N \end{bmatrix}}_{:=\mathbf{V}_{Cabc}^{PN}} \approx \underbrace{\begin{bmatrix} p_a^P & p_b^P & p_c^P \\ p_a^N & p_b^N & p_c^N \end{bmatrix}}_{:=\mathbf{P}_{abc}^{PN} = \mathbf{V}_{abc}^{PN} \circ \mathbf{I}_{abc}^{PN}} \tag{5}$$

where "$\circ$" denotes the element-by-element multiplication of two matrices and $v_{Ca}^P = \sum_{k=1}^{n} v_{Cak}^P$, etc., are the cluster voltages. In (5), it is supposed that the capacitor voltages are maintained around a quiescent value, $\bar{v}_C$, with relatively small ripple. Applying the $\Sigma\Delta\alpha\beta0$-transformation to (5) yields

$$C\bar{v}_C \frac{d}{dt} \underbrace{\begin{bmatrix} v_{C\alpha}^\Sigma & v_{C\beta}^\Sigma & v_{C0}^\Sigma \\ v_{C\alpha}^\Delta & v_{C\beta}^\Delta & v_{C0}^\Delta \end{bmatrix}}_{:=\mathbf{V}_{C\alpha\beta0}^{\Sigma\Delta}} \approx \underbrace{\begin{bmatrix} p_\alpha^\Sigma & p_\beta^\Sigma & p_0^\Sigma \\ p_\alpha^\Delta & p_\beta^\Delta & p_0^\Delta \end{bmatrix}}_{:=\mathbf{P}_{\alpha\beta0}^{\Sigma\Delta}} \tag{6}$$

As discussed in previous publications [5], the vector model of (6) is more appropriate for variable-speed applications. Defining the variables as vectors (e.g., $\mathbf{i}_{\alpha\beta} = i_\alpha + ji_\beta$), the vector model of (6) results in [5]

$$C\bar{v}_C \frac{d\mathbf{v}_{C\alpha\beta}^\Sigma}{dt} \approx \frac{1}{2}E\mathbf{i}_{\alpha\beta}^\Sigma - \frac{1}{4}\left(\mathbf{i}_{\alpha\beta}\mathbf{v}_{\alpha\beta}\right)^c - \frac{1}{2}v_0\mathbf{i}_{\alpha\beta} \tag{7}$$

$$C\bar{v}_C \frac{d\mathbf{v}_{C\alpha\beta}^\Delta}{dt} \approx \frac{1}{2}E\mathbf{i}_{\alpha\beta} - \frac{2}{3}i^P\mathbf{v}_{\alpha\beta} - \left(\mathbf{v}_{\alpha\beta}\mathbf{i}_{\alpha\beta}^\Sigma\right)^c - 2v_0\mathbf{i}_{\alpha\beta}^\Sigma \tag{8}$$

$$C\bar{v}_C \frac{dv_{C0}^\Sigma}{dt} \approx \frac{1}{6}Ei^P - \frac{1}{4}\Re\left[\mathbf{v}_{\alpha\beta}(\mathbf{i}_{\alpha\beta})^c\right] \tag{9}$$

$$C\bar{v}_C \frac{dv_{C0}^\Delta}{dt} \approx -\Re\left[\mathbf{v}_{\alpha\beta}(\mathbf{i}_{\alpha\beta}^\Sigma)^c\right] - \frac{2}{3}i^P v_0 \tag{10}$$

where "$c$" represents the complex conjugated operator. An advantage of the vector model (7)–(10) is that the amplitude of the oscillations in the cluster voltages can be approximated considering that only the vectors $\mathbf{v}_{C\alpha\beta}^\Delta$ and $\mathbf{v}_{C\alpha\beta}^\Sigma$ present important voltage fluctuations. For instance, using the inverse $\Sigma\Delta\alpha\beta0$-transformation yields (see Figure 2)

$$|\tilde{v}_{Cx}^X| \le \frac{1}{2}|\mathbf{v}_{C\alpha\beta}^\Delta| + |\mathbf{v}_{C\alpha\beta}^\Sigma| = \frac{1}{2}|\mathbf{v}_{Cdq}^\Delta| + |\mathbf{v}_{Cdq}^\Sigma| \tag{11}$$

Notice that $|\tilde{v}_{Cx}^X|$ is calculated in the $\alpha\beta$- and $dq$-coordinated systems, and it has to be within an acceptable value to ensure the proper operation of the MMC.

As reported in previous works [5,6], the unacceptably high values of $|\tilde{v}_{Cx}^X|$ are produced by $\mathbf{v}_{Cdq}^\Delta$ during low machine frequencies. Representing (8) in a $dq$-coordinated system rotating at $\omega_e$ yields

$$C\bar{v}_C \frac{d\mathbf{v}_{Cdq}^\Delta}{dt} \approx \underbrace{\frac{1}{2}E\mathbf{i}_{dq} - \frac{2}{3}i^P\mathbf{v}_{dq}}_{:=\mathbf{p}_\omega} - \underbrace{jC\bar{v}_C\omega_e\mathbf{v}_{Cdq}^\Delta}_{:=\mathbf{p}_m} - \underbrace{2v_0\mathbf{i}_{dq}^\Sigma}_{:=\mathbf{p}_c} \tag{12}$$

where the term $(\mathbf{v}_{\alpha\beta}\mathbf{i}_{\alpha\beta}^\Sigma)^c$ in (8) is neglected since it does not produce important variations in $\mathbf{v}_{Cdq}^\Delta$. Notice that any low-frequency component in the right side of (12) has to be eliminated; otherwise, $\mathbf{v}_{Cdq}^\Delta$ and $|\tilde{v}_{Cx}^X|$ will increase dramatically. Therefore, the LFM must be enabled when $|\mathbf{p}_\omega| > |\mathbf{p}_m|$, because the vector $\mathbf{p}_c = 2v_0\mathbf{i}_{dq}^\Sigma$ is required to ensure the stability of $\mathbf{v}_{Cdq}^\Delta$. On the other hand, the HFM is enabled when $\mathbf{p}_\omega \approx \mathbf{p}_m$, and then $\mathbf{p}_c$ can be set to approximately **0**.

Equation (12) can be also utilized to describe the control strategies discussed in Section 1 and then the behavior of $|\tilde{v}_{Cx}^X|$. For example, the full mitigationmethod (see Figure 2a) is used when $\mathbf{v}_{Cdq}^\Delta$ is completely eliminated during the whole LFM, requiring large circulating currents because $\mathbf{p}_c$ has to eliminate $\mathbf{p}_\omega$. However, when $\mathbf{v}_{Cdq}^\Delta$ is regulated to a non-zero value (i.e., the margin-based method is implemented; see Figure 2b), both vectors $\mathbf{p}_c$ and $\mathbf{p}_m$ are used to eliminate $\mathbf{p}_\omega$, reducing the amplitude of the circulating currents in $\mathbf{p}_c$. Finally, when the variable DC port voltage method is used (see Figure 2c), $\mathbf{p}_\omega$ is reduced by varying $E$, and then both vectors $\mathbf{p}_c$ and $\mathbf{p}_m$ are allowed to have a lower amplitude.

## 3. Proposed Modeling and Design Approach

### 3.1. Influence of the DC Port Voltage on the MMC Behavior

Based on (11) and (12), it is possible to define a state-space model to relate the amplitude of the oscillations in the cluster voltages ($|\tilde{v}_{Cx}^X|$) to the DC port voltage ($E$). The proposed model can be then used to configure a proper controller that manipulates $E$ to regulate $|\tilde{v}_{Cx}^X|$, as shown below.

To obtain the proposed model, $v_{Cd}^\Delta$ and $v_{Cq}^\Delta$ are solved from (12), setting $\mathbf{i}_{dq}^\Sigma \approx \mathbf{0}$, because it is supposed that the converter operates in HFM, as follows:

$$C\bar{v}_C \frac{dv_{Cd}^\Delta}{dt} \approx \tfrac{1}{2}Ei_d - \tfrac{2}{3}i^P v_d + C\bar{v}_C \omega_e v_{Cq}^\Delta \Rightarrow \frac{dv_{Cd}^\Delta}{dt} \approx \omega_e v_{Cq}^\Delta + \frac{i_d}{2C\bar{v}_C}E - \frac{2p_o v_d}{3C\bar{v}_C}\frac{1}{E} \tag{13}$$

$$C\bar{v}_C \frac{dv_{Cq}^\Delta}{dt} \approx \tfrac{1}{2}Ei_q - \tfrac{2}{3}i^P v_q - C\bar{v}_C \omega_e v_{Cd}^\Delta \Rightarrow \frac{dv_{Cq}^\Delta}{dt} \approx -\omega_e v_{Cd}^\Delta + \frac{i_q}{2C\bar{v}_C}E - \frac{2p_o v_q}{3C\bar{v}_C}\frac{1}{E} \tag{14}$$

where $p_o$ is the machine power and $i^P \approx \frac{p_o}{E}$ (i.e., the converter losses are low). Notice that (13) and (14) define two dynamic expressions for $v_{Cd}^\Delta$ and $v_{Cq}^\Delta$ as a non-linear function of $E$ (see the last term of (13) and (14)). To relate these two expressions to $|\tilde{v}_{Cx}^X|$, (11) is considered since $|\mathbf{v}_{Cdq}^\Delta| = \sqrt{(v_{Cd}^\Delta)^2 + (v_{Cq}^\Delta)^2}$ and $|\mathbf{v}_{Cdq}^\Sigma|$ can be calculated by supposing that $\mathbf{i}_{dq}^\Sigma \approx \mathbf{0}$ in (7), applying the *dq*-transformation and integrating them, resulting in

$$|\tilde{v}_{Cx}^X| \approx \frac{1}{2}\underbrace{\sqrt{(v_{Cd}^\Delta)^2 + (v_{Cq}^\Delta)^2}}_{|\mathbf{v}_{Cdq}^\Delta|} + \underbrace{\frac{|\mathbf{i}_{dq}||\mathbf{v}_{dq}|}{8C\bar{v}_C\omega_e}}_{|\mathbf{v}_{Cdq}^\Sigma|} \tag{15}$$

In this manner, a non-linear dynamic model between $|\tilde{v}_{Cx}^X|$ and $E$ can be defined using (13)–(15) as

$$\frac{dv_{Cd}^\Delta}{dt} \approx \omega_e v_{Cq}^\Delta + \frac{i_d}{2C\bar{v}_C}E - \frac{2p_o v_d}{3C\bar{v}_C}\frac{1}{E} \tag{16}$$

$$\frac{dv_{Cq}^\Delta}{dt} \approx -\omega_e v_{Cd}^\Delta + \frac{i_q}{2C\bar{v}_C}E - \frac{2p_o v_q}{3C\bar{v}_C}\frac{1}{E} \tag{17}$$

$$|\tilde{v}_{Cx}^X| \approx \frac{1}{2}\sqrt{(v_{Cd}^\Delta)^2 + (v_{Cq}^\Delta)^2} + \frac{|\mathbf{i}_{dq}||\mathbf{v}_{dq}|}{8C\bar{v}_C\omega_e} \tag{18}$$

The previously defined model can be linearized around an operating point "0". Using the operator "$\delta$" to denote small changes around the point of operation, the linear state-space model is obtained from (16)–(18) as follows:

$$\frac{d}{dt}\begin{bmatrix} \delta v_{Cd}^\Delta \\ \delta v_{Cq}^\Delta \end{bmatrix} \approx \begin{bmatrix} 0 & \omega_{e0} \\ -\omega_{e0} & 0 \end{bmatrix}\begin{bmatrix} \delta v_{Cd}^\Delta \\ \delta v_{Cq}^\Delta \end{bmatrix} + \begin{bmatrix} \frac{i_{d0}}{2C\bar{v}_{C0}} + \frac{2p_{o0}v_{d0}}{3C\bar{v}_{C0}E_0^2} \\ \frac{i_{q0}}{2C\bar{v}_{C0}} + \frac{2p_{o0}v_{q0}}{3C\bar{v}_{C0}E_0^2} \end{bmatrix}\delta E \tag{19}$$

$$\delta|\tilde{v}_{Cx}^X| \approx \begin{bmatrix} \dfrac{v_{Cd0}^\Delta}{2|\mathbf{v}_{Cdq0}^\Delta|} & \dfrac{v_{Cq0}^\Delta}{2|\mathbf{v}_{Cdq0}^\Delta|} \end{bmatrix} \begin{bmatrix} \delta v_{Cd}^\Delta \\ \delta v_{Cq}^\Delta \end{bmatrix} \tag{20}$$

considering that the operating point of $v_{Cd}^\Delta$ and $v_{Cq}^\Delta$ is obtained from (16) and (17), the transfer function from the small changes in the DC port voltage, $\delta E$, to the small changes in the amplitude of the cluster voltage fluctuations, $\delta|\tilde{v}_{Cx}^X|$, is determined from (19) and (20) as

$$\frac{\delta|\tilde{v}_{Cx}^X|(s)}{\delta E(s)} = \frac{k_1 s + k_2}{s^2 + \omega_{e0}^2} \tag{21}$$

where $q_{o0}$ is the reactive power of the electrical machine and

$$k_1 = \frac{-4 p_{o0} q_{o0}}{3 C \bar{v}_C \sqrt{9 E_0^4 |\mathbf{i}_{dq0}|^2 + 16 p_{o0}^2 \left( |\mathbf{v}_{dq0}|^2 - E_0^2 \right)}}, \tag{22}$$

$$k_2 = \frac{9 E_0^4 |\mathbf{i}_{dq0}|^2 \omega_{e0} - 16 p_{o0}^2 |\mathbf{v}_{dq0}|^2 \omega_{e0}}{12 E_0^2 C \bar{v}_C \sqrt{9 E_0^4 |\mathbf{i}_{dq0}|^2 + 16 p_{o0}^2 \left( |\mathbf{v}_{dq0}|^2 - E_0^2 \right)}}, \tag{23}$$

From the proposed model given in (21), it is concluded that sudden changes in $E$ are likely to cause fluctuations in frequency $\omega_e$ in $|\tilde{v}_{Cx}^X|$ due to its poles ($\pm j\omega_{e0}$), indicating a poor damping factor in the non-linear model (13)–(15). These types of oscillations jeopardize the quality of the output voltage of the MMC, decrease the lifespan of the converter capacitors, and produce disturbances in the voltage-balancing controllers of the MMC.

From the proposed model given in (21), it is concluded that sudden changes in $E$ are
As a representative example, Figure 3 shows the values of the parameters of (21) as a function of $\omega_e$. The drive and machine parameters are shown in Table 1. The objective of this simulation is to maintain $|\tilde{v}_{Cx}^X|$ equal to its value achieved at nominal power and speed. The DC port voltage was obtained from equalling $|\mathbf{p}_\omega| = |\mathbf{p}_m|$, setting $|\mathbf{v}_{Cdq}^\Delta| \approx 2(|\tilde{v}_{Cx}^X| - |\mathbf{v}_{Cdq}^\Sigma|)$, and solving $E$ (see (11) and Figure 3a). The following per-unit load torque was applied:

$$\tau_e(\omega_{e(\text{p.u.})}) = 0.2 + 0.8 \omega_{e(\text{p.u.})}^2 \tag{24}$$

As shown in Figure 3b, the non-minimum phase zero of (21) is dominant for low frequencies. This zero is proportional to the reactive power of the machine; thus, it cannot be presented in synchronous machines (see (22)). Figure 3c depicts the maximum value of $\delta|\tilde{v}_{Cx}^X|$ for a unit step in $\delta E$, which is higher as the machine frequency decreases and approximately 25% of $\delta E$ for medium and close to nominal frequencies. Therefore, large changes in the DC port voltage could jeopardize the performance of the system due to changes in $|\tilde{v}_{Cx}^X|$.

**Table 1.** Set-up parameters for the 18-cell MMC drive.

| Parameter | Symbol | Value | Unit |
|---|---|---|---|
| Nominal DC port voltage | $E$ | 300 | V |
| Cluster inductor | $L$ | 2.5 | mH |
| Cell capacitor | $C$ | 4700 | μF |
| Cell DC voltage | $\bar{v}_C^*$ | 100 | V |
| Cell count per cluster | $n$ | 3 | – |

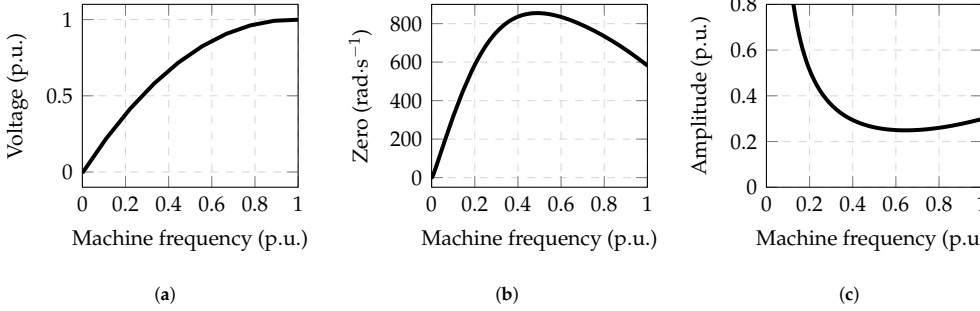

**Figure 3.** Parameters of (21) for $|\tilde{v}_{Cx}^X| = $ const. (**a**) Required DC port voltage; (**b**) zero value; (**c**) maximum value.

*3.2. Restrictions in the Variation of the DC Port Voltage*

Figure 3a also shows that low values of $E$ must be used to maintain a constant $|\tilde{v}_{Cx}^X|$ during low machine frequencies. Although the common-mode voltage decreases as $E$ reduces and then the winding insulation damages and leakage currents in the bearing are reduced [20,21], the use of very low values of $E$ is not recommended in this work due to the following.

1. The MMC clusters have to synthesize positive voltages with a mean value of $\frac{E}{2}$; thus, very low values of $E$ compromise the quality of the output voltage and then the broader controllability of the MMC.
2. As the input power must always be ensured, large values of the DC port current ($i^P$) could result if $E$ is severely reduced, which could increase the conduction losses or cause the oversizing of the converter.
3. Small changes in $E$ produce large changes in $|\tilde{v}_{Cx}^X|$ when the converter operates at very low frequencies (see Figure 3c). Therefore, the sensibility of the control system could be compromised.

Consequently, the DC port voltage of the MMC cannot be arbitrarily low and another procedure has to be used to regulate $|\tilde{v}_{Cx}^X|$ as desired during low frequencies.

To achieve the aforementioned requirement, this work analyzes the application of the margin-based method presented in [2,9] when $E$ is reduced but not equal to 0 during LFM. In this method, the power $\mathbf{p}_c = 2v_0\mathbf{i}_{dq}^\Sigma$ eliminates the power fluctuations $\mathbf{p}_\omega$ and $\mathbf{p}_m$ in (12). In this case, the set-point values of $\tilde{\mathbf{i}}_{dq}^\Sigma$ and $\tilde{v}_0$ are defined in [2,9] as

$$\tilde{\mathbf{i}}_{dq}^{\Sigma*} = \frac{1}{2V_0}(\mathbf{p}_\omega - \mathbf{p}_m)f(t), \quad \tilde{v}_0^* = V_0\, g(t) \tag{25}$$

where "*" stands for the desired value of a variable, $V_0$ is the amplitude of the common-mode voltage, "~" is the AC part of a variable, and the functions $g(t)$ and $f(t)$ are two high-frequency signals such that the mean value of $g(t)f(t)$ is one. Notice that an additional current vector can be added to $\tilde{\mathbf{i}}_{dq}^{\Sigma*}$ to ensure the regulation of $\mathbf{v}_{Cdq}^\Delta$ (see Section 4). Additionally, $\tilde{\mathbf{i}}_{dq}^{\Sigma*}$ is minimized when $\mathbf{p}_\omega$ and $\mathbf{p}_m$ are in-phase. Therefore, using (15), the set-point value of the vector $\mathbf{v}_{Cdq}^\Delta$ results in

$$\mathbf{v}_{Cdq}^{\Delta*} = -j2\mathrm{sign}[\omega_e] \underbrace{\left( |\tilde{v}_{Cx}^X|^* - |\mathbf{v}_{Cdq}^\Sigma| \right)}_{|\mathbf{v}_{Cdq}^\Delta|^*} \hat{\mathbf{p}}_\omega \tag{26}$$

where $\hat{\mathbf{p}}_\omega$ is an unitary vector in-phase with $\mathbf{p}_\omega$. Consequently, $\mathbf{v}_{Cdq}^\Delta$ is controlled to ensure the desired value of $|\tilde{v}_{Cx}^X|^*$, which is the objective of the margin-based method [2,9].

Once the margin-based method has been introduced, the influence of $E$ in the mitigating variables is analyzed. Notice that, neglecting the inductor voltage drop, the amplitude of the common-mode voltage is given by [3]:

$$V_0 \approx \frac{1}{2}E - |\mathbf{v}_{dq}| \Rightarrow \max[V_0] = \frac{1}{2}E \tag{27}$$

As indicated in [21], the high-frequency common-mode voltage causes a leakage current between the machine frame and the bearings, because there is a stray capacitor present between them (see [20] for a detailed model of this phenomenon). This leakage current may damage the motor bearings, increasing the maintenance costs and production interruptions. Therefore, as $\max[V_0]$ is proportional to $E$, low values of $E$ during LFM mitigate these problems and increase the whole system's effectiveness, feasibility, and robustness.

On the other hand, the circulating currents are maximum at the machine's start-up when the margin-based strategy is applied [2,9,10]. In this condition, $\mathbf{p}_\omega \approx \frac{1}{2}E\mathbf{i}_{dq}$ and $\mathbf{p}_m = jC\bar{v}_C\omega_e\mathbf{v}_{Cdq}^\Delta \approx \mathbf{0}$. Therefore, $\max[\mathbf{i}_{dq}^\Sigma]$ results in

$$\max[\mathbf{i}_{dq}^\Sigma] \approx \max[\mathbf{i}_{dq}f(t)], \tag{28}$$

and, then, the maximum value of the circulating currents is not affected by changes in $E$. Therefore, it can be concluded that the reduction in $E$ does not alter negatively the mitigation variables required in the LFM, and then the margin-based method can be applied to regulate $|\tilde{v}_{Cx}^X|$. Moreover, the common-mode voltage is reduced in the same proportion as $E$, which is beneficial from the machine point of view.

### 3.3. Design Procedure

Summarizing the analyses of the previous sections, the following procedure is presented to design an MMC-based drive with a variable DC port voltage.

1. Given a machine and application, determine $\mathbf{i}_{dq}$, $\mathbf{v}_{dq}$, and $p_o$ as a function of $\omega_e$.
2. According to the maximum value of $\mathbf{v}_{dq}$, design the number of cells ($n$), the mean value of the capacitor voltages ($\bar{v}_C$), and the nominal DC port voltage ($E_{\text{nom}}$).
3. Define a minimum value of $E$, $E_{\min}$, and compute $|\mathbf{p}_\omega(E_{\min})|$ as follows (see (12)):

$$|\mathbf{p}_\omega(E_{\min})| = \left| \frac{1}{2}E_{\min}\mathbf{i}_{dq}(\omega_e) - \frac{2}{3}\frac{p_o(\omega_e)}{E_{\min}}\mathbf{v}_{dq}(\omega_e) \right| \tag{29}$$

   $E_{\min}$ must be high enough to synthesize the output voltage required by the clusters.
4. By inserting (15) into (12), determine $|\mathbf{p}_m|$ as follows:

$$|\mathbf{p}_m| = 2C\bar{v}_C\omega_e|\tilde{v}_{Cx}^X| - \frac{1}{4}|\mathbf{i}_{dq}||\mathbf{v}_{dq}| \tag{30}$$

5. Define $|\tilde{v}_{Cx}^X|^* = |\tilde{v}_{Cx}^X|$. Notice that $|\mathbf{p}_m|$ remains constant if the product $C|\tilde{v}_{Cx}^X|^*$ is also constant (see (30)), reflecting the trade-off between the cell capacitance and the voltage fluctuations in the capacitors.
6. Plot $|\mathbf{p}_\omega(E_{\min})|$ and $|\mathbf{p}_m|$ and determinate the intersection point (see the solid red and black lines in Figure 4a, which is based on the system described in Table 1). This point is the nominal transition between the LFM and HFM, which could be modified by changing the product $C|\tilde{v}_{Cx}^X|$ or the value of $E_{\min}$. Notice that $|\mathbf{p}_m| = |\mathbf{p}_\omega|$ after the transition point (see the dashed red line of Figure 4a).
7. Solve $|\mathbf{p}_m| = |\mathbf{p}_\omega|$ for $E$. The blue line in Figure 4b shows the DC port voltage obtained in this example. Notice that the low limit $E$ is $E_{\min}$ (see the gray line).
8. Compute the model (21) by using the information of the system variables and tune a proper controller to regulate $|\tilde{v}_{Cx}^X|$. The value of $E$ of the previous step can be stored in a look-up table and used for feed-forward purposes. Considering the low damping

and the changes in the model parameters, algorithms for active damping or robust tuning rules can be applied, as presented in the next section.

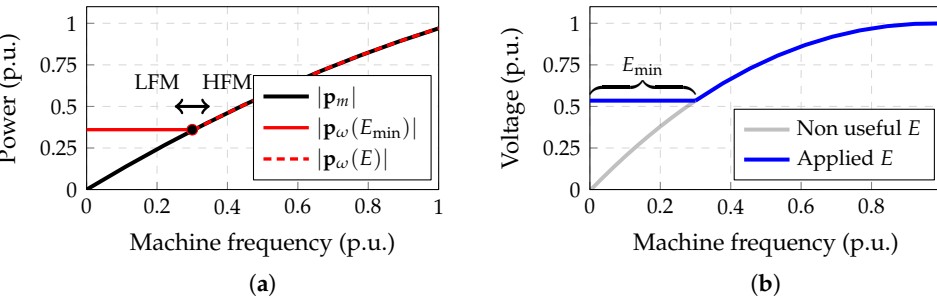

**Figure 4.** Design procedure of the MMC-based drive. (**a**) Power flows $\mathbf{p}_\omega$ and $\mathbf{p}_m$; (**b**) DC port voltage $E$.

## 4. Proposed Control System

The proposed control system is shown in Figures 5 and 6. The white arrows indicate vector signals; meanwhile, the scalar ones are denoted by black arrows. For simplicity, the feed-forward terms are not presented. As indicated in Figure 5, the amplitude of the voltage fluctuation in the capacitors ($|\tilde{v}_{Cx}^X|$) is kept constant using two different methods.

1. During LFM, the green blocks in Figure 5 use the margin-based method to keep them constant (i.e., by injecting circulating currents). In this mode, the PI controller that manipulates the DC port voltage $E^*$ is saturated to its minimum value $E_{min}^*$.

2. When the system operates in HFM, this controller is not saturated and it modifies $E^*$ to ensure the desired value of the voltage fluctuation in the capacitors, considering the dynamic relation between them and the DC port voltage (see Section 3.1 and Equation (21)).

Notice that during LFM, the DC port voltage is reduced to its minimum feasible value; then, the applied common-mode voltage is also minimized. As discussed in Section 3.2, this reduction protects the machine bearings, increasing the system's feasibility and robustness.

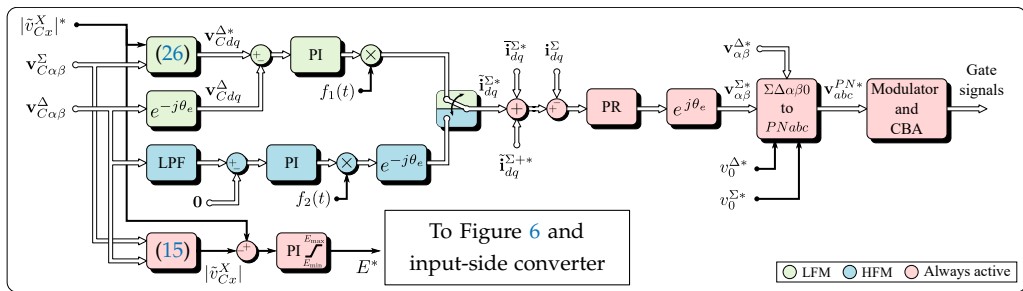

**Figure 5.** Proposed control system to regulate $|\tilde{v}_{Cx}^X|$, $\mathbf{v}_{C\alpha\beta}^\Sigma$ and the circulating currents.

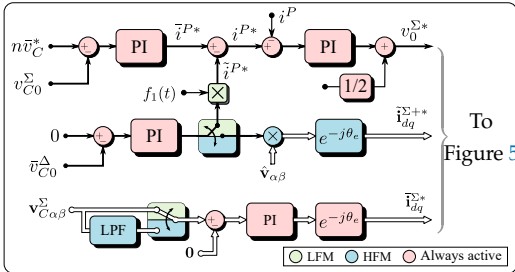

**Figure 6.** Proposed control system to control $\mathbf{v}_{C\alpha\beta}^\Sigma$, $v_{C0}^\Delta$ and $v_{C0}^\Sigma$.

Considering the high number of parameters in the process (see (21)), optimal tuning for the PI controller is used. The tuning criteria consider the performance/robustness trade-off of the control loop by solving the following optimization problem:

$$\min_{K_p, K_i} \sum_{h=0}^{\infty} \left| |\tilde{v}_{Cx}^X|^* - |\tilde{v}_{Cx}^X| \right| \text{ s.t.: } M_S^* = 2 \tag{31}$$

where $M_S$ is the peak of the sensitivity function. It is common to consider $M_S \leq 2$ as a good robustness indicator, resulting in $A_m \geq 2$ and $\phi_m \geq 28.9°$ [23]. Thus, the system's stability is guaranteed for changes in the nominal model of the process or changes in its parameters, as expected in (21). Notice that the value of $E$ is sent to the grid-side converter, which is not covered in this paper due to space limitations (see [19]).

During the LFM (see the green blocks in Figure 5), two PI controllers are applied to regulate $\mathbf{v}_{Cdq}^\Delta$ as defined by $\mathbf{v}_{Cdq}^{\Delta*}$ (see (26)). The output of these PI controllers is multiplied by the high-frequency function $f_1(t)$, producing a pulsating-set-point value of the circulating currents, $\tilde{\mathbf{i}}_{dq}^{\Sigma*}$, in-phase with the common-mode voltage $v_0(t)$. During the LFM, the common-mode voltage is usually imposed as a square waveform (see [5] and (25)).

Additionally, during the HFM (see the light blue blocks in Figure 5), the AC component of $\mathbf{v}_{C\alpha\beta}^\Sigma$ is filtered out using low-pass filters (LPF) and regulated to zero by two PI controllers, ensuring the balance of $\mathbf{v}_{C\alpha\beta}^\Sigma$. Additionally, the output of these controllers is multiplied by $f_2(t)$ and transformed into $dq$-coordinates to generate $\tilde{\mathbf{i}}_{dq}^{\Sigma*}$ (the common-mode voltage, as well as function $f_2(t)$, are defined as in-phase third-harmonic waves to increase the modulation index). Notice that $\tilde{\mathbf{i}}_{dq}^{\Sigma*}$ is chosen depending on the MMC mode, i.e., LFM if $|\mathbf{p}_\omega| > \mathbf{p}_m$ and HFM if $|\mathbf{p}_\omega| < \mathbf{p}_m$. A hysteresis band can be implemented to avoid repetitive transitions due to noise in $|\mathbf{p}_\omega|$ and $\mathbf{p}_m$.

Once $\tilde{\mathbf{i}}_{dq}^{\Sigma*}$ is defined, other components imposed by the control system shown in Figure 6 are added to define the final value of $\mathbf{i}_{dq}^{\Sigma*}$, which is ensured using proportional-resonant controllers. The same tuning procedure of (31) can be used to tune these controllers. The output of the resonant controllers is then transformed from $\alpha\beta$- to $dq$-coordinates and used to define the component $\mathbf{v}_{\alpha\beta}^{\Sigma*}$ of the desired output cluster voltages. The other components are defined by the control system shown in Figure 6 and the machine control system (see [24]). However, it is important to mention that the desired common-mode voltage can be directly imposed by modifying the voltage $v_0^\Delta$ as follows (see (4)):

$$v_0^\Delta = -2v_0 \Rightarrow v_0^{\Delta*} = -2v_0^* \tag{32}$$

Therefore, the difference between the desired value of the common-mode voltage and the obtained one is only due to the modulation algorithm, which is neglected in most of the converters, ensuring the proper injection of the common-mode voltage, as discussed in this paper in (25) and (27). Finally, the output cluster voltages are transformed into natural coordinates and sent to the modulator and the cell-balancing algorithm (CBA in Figure 5).

As depicted in Figure 6, the regulation of $\mathbf{v}_{C\alpha\beta}^\Sigma$ is realized by adding a DC component to the circulating currents, to manipulate the power flow $\frac{1}{2}E\mathbf{i}_{\alpha\beta}^\Sigma$ (see (7)). As shown in the green blocks of Figure 6, $\mathbf{v}_{C\alpha\beta}^\Sigma$ is directly regulated during the LFM. However, when the switches enable the HFM (see the light blue blocks of Figure 6), only the low-frequency components of $\mathbf{v}_{C\alpha\beta}^\Sigma$ are regulated, avoiding the presence of circulating currents in steady state.

Finally, the voltage $v_{C0}^\Sigma$ is regulated using a DC component in $i^P$, as shown at the top of Figure 6 (see (9)). On the other hand, the voltage $v_{C0}^\Delta$ (see (10)) can be regulated using (1) an AC component in $i^P$, which produces a non-zero mean power with the common-mode voltage, or (2) a component of the circulating currents in-phase with the machine back EMF (see (10)). Considering that during the LFM, the common-mode voltage is high, the first option is used in this mode (see the green blocks of Figure 6). The second option is utilized

during the HFM considering that the back EMF is high enough in this operating mode (see the light blue blocks in Figure 6).

## 5. Experimental Results

Experimental tests have been developed using a BTB MMC-based drive (see Figure 7). A photograph of the system is shown in Figure 7a. The grid-side MMC is composed of 12 power cells and operates in boost mode. The parameters of the machine-side converter are given in Table 1. As shown in Figure 7b, the MMC drives a 7.5 kW induction machine connected to a permanent magnet generator (PMG) with a resistor bank as an electrical load. Each MMC is controlled using a DSP model TMS320C6713 and 2 Actel ProAsic3 FPGA boards. A phase-shifted PWM algorithm generates the switching signals timed by the FPGA. An optical fiber transmits the switching signals to the gate drivers of the MOSFET (model IRFP4868PbF). An SPI communication link has been implemented to enable communication between the control platforms of both MMCs and send the desired value of $E$ to the grid-side converter.

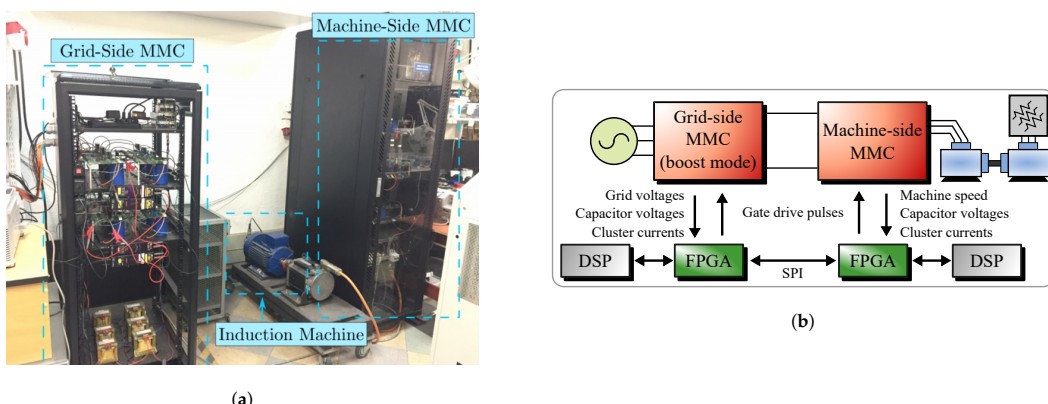

**Figure 7.** Experimental set-up description. (**a**) Experimental system; (**b**) connection diagram.

### 5.1. Steady-State Comparison with a Conventional Method

Experimental tests have been carried out to compare the influence of the DC port voltage on the mitigating variables $v_0$ and $\mathbf{i}_{dq}^{\Sigma}$ (see Figures 8 and 9). In the tests, the machine rotor has been locked and then $|\mathbf{p}_m| \approx 0$, producing the maximum circulating currents and common-mode voltages for both cases (see (12)).

For the experimental test depicted in Figure 8a, the DC port voltage $E$ is regulated to 300 V; meanwhile, for the experimental work depicted in Figure 8b, the DC port voltage $E$ is 150 V. In each of the scope shots, from top to bottom, the first waveform corresponds to the output cluster voltage ($v_a^P$), the second ones are the cluster currents ($i_a^P$, $i_a^N$), the third one is the circulating current in one phase (calculated as $i_a^{\Sigma} = \frac{1}{2}(i_a^P + i_a^N)$), and the last waveform corresponds to the machine stator current ($i_a$). As discussed in Section 3.2, the common-mode voltage is reduced in the same proportion as the DC port voltage, and the magnitude of the circulating current is not notably modified. This is corroborated by comparing the cluster voltages and the waveforms of the circulating currents produced by both experimental tests.

Although it is not possible to show the experimental waveform of the DC port voltage because the machine is delta-connected, indirect measurements have been carried out to demonstrate the effectiveness of the proposed method to reduce this voltage. As is well known, the cluster voltage $v_a^P$ can be written as a function of other voltages as follows (see (1)):

$$v_a^P = -L\frac{di_a^P}{dt} + \frac{E}{2} - v_a \Rightarrow v_a^P = -L\frac{di_a^P}{dt} + \frac{E}{2} - (v_a' + v_0) \approx \frac{E}{2} - v_0 \qquad (33)$$

where the term $L\frac{di_a^P}{dt}$ is neglected because of the low value of the inductance, and $v_a'$ is the phase *a* voltage, free of any common-mode voltage. Consequently, $v_a'$ can be also neglected during rotor-locked conditions. From (33), it can be concluded that the harmonics of $v_a^P$ have to be principally conformed by a DC term imposed by the DC port voltage, as well as a term of the same frequency of the common-mode voltage, which corresponds to the mitigating frequency of the LFM.

In Figure 9, the results of the Fast Fourier Transform (FFT) applied to the cluster voltages achieved with both experimental test are depicted. The black line corresponds to $E = 300$ V. For the rotor-locked test, the amplitude of the common-mode voltage is the maximum (i.e., ≈137 V), and it has a fundamental frequency of 75 Hz, as shown in Figure 9 (black line). The red line in Figure 9 corresponds to $E = 150\ V$. Notice that there is a harmonic in 75 Hz with an amplitude of 63 V, which corresponds to the common-mode voltage, which has been reduced to approximately half of the previous value, and this is in broad agreement with the analysis presented in Section 3.2 and Equation (27), where the common-mode voltage is defined as a function of the applied DC port voltage.

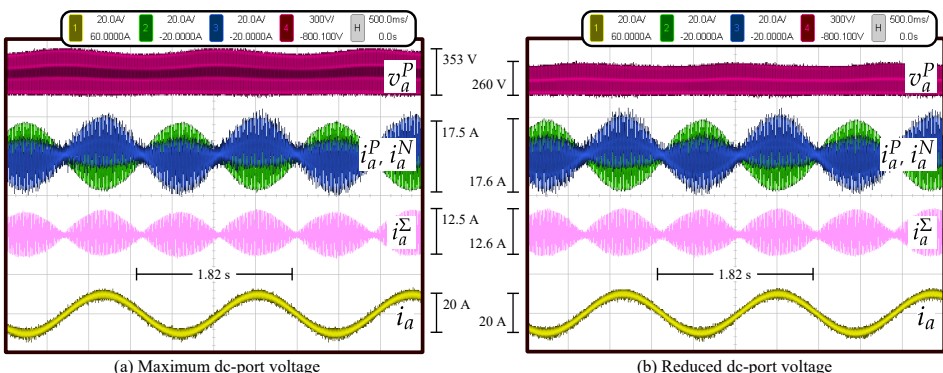

(a) Maximum dc-port voltage        (b) Reduced dc-port voltage

**Figure 8.** Locked rotor performance. (**a**) $E = E_{\max}$. (**b**) $E = 0.5E_{\max}$. Output cluster voltage $v_a^P$, cluster currents $i_a^P$ and $i_a^N$, circulating current $i_a^\Sigma = \frac{1}{2}(i_a^P + i_a^N)$, machine current $i_a$.

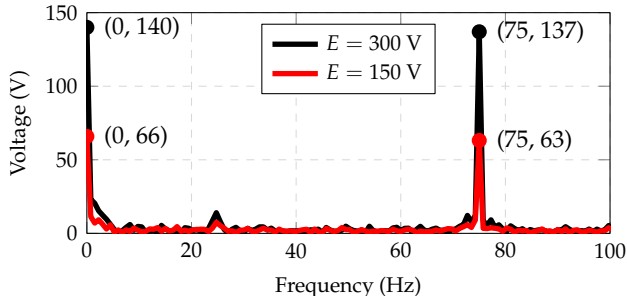

**Figure 9.** Fourier Transform of the cluster voltage of Figure 8. Black line: $E = 300$ V, red line: $E = 150$ V.

### 5.2. Experimental Comparison of the Dynamic Performance

Figure 10 depicts the experimental results when comparing the conventional margin-based strategy with the nominal DC port voltage [2,9] (left side) and the proposed control scheme (right side). During the test, the machine was accelerated from 0 to 1400 rpm in 15 s. A margin of ±10% of the mean value of the cluster voltages was chosen for the margin-based strategy (i.e., $|\tilde{v}_{Cx}^X|^* = 30$ V). All the data were obtained by sampling the experimental waveforms with the FPGA and DSP platforms described at the beginning of Section 5.

The obtained machine speed and currents are shown in Figure 10a,b and Figure 10c,d, respectively. Notice that these variables are essentially the same for both control strategies, indicating that there is no difference in the machine conditions. The applied DC port voltage (red signal) and the output cluster voltage (black signal) are shown in Figure 10e,f.

As shown in the figures, a higher output cluster voltage is required during LFM when the nominal DC port voltage is used (compare Figure 10e with Figure 10f). Additionally, the mitigating common-mode voltage is reflected in the output cluster voltage, indicating the duration of the LFM for both cases (approximately 30% of the total operating range, which is considered a reasonable value [6]).

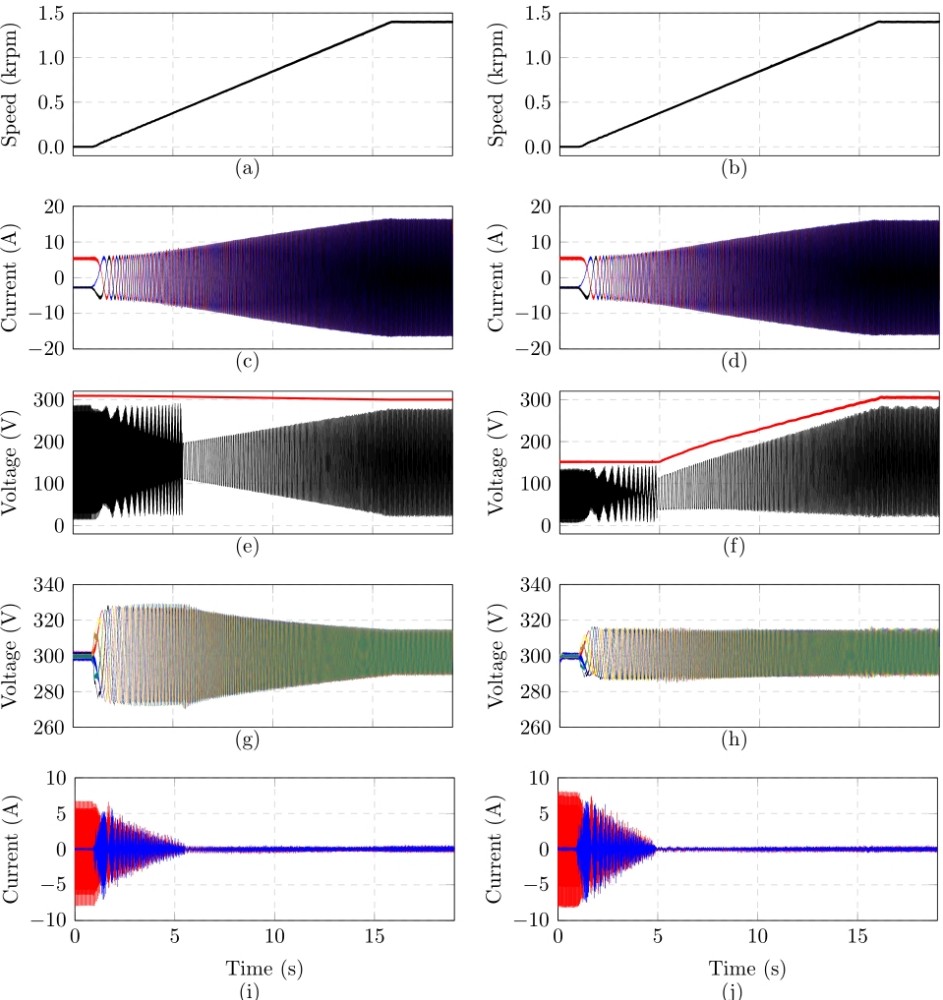

**Figure 10.** Comparison of strategies. (**Left**) Margin-based strategy. (**Right**) Proposed strategy. (**a,b**) Machine speed, (**c,d**) machine currents, (**e,f**) DC port voltage, (**g,h**) cluster voltages, (**i,j**) circulating currents.

As expected in the conventional margin-based strategy, the cluster voltages are maintained within the voltage margin defined by $|\tilde{v}_{Cx}^{X}|^{*} = 30$ V, but their fluctuations decrease until approximately a 17 V peak during HFM (see Figure 10g). In the case of the proposed control strategy (see Figure 10h), the cluster voltages are always within the margin defined by $|\tilde{v}_{Cx}^{X}|^{*} = 17$ V during the whole operating range, which demonstrates the effectiveness of the proposed control scheme. As depicted in Figure 10i,j, the mitigating currents are not severely affected by the variations in the DC port voltage. When a constant voltage $E$ is utilized (see Figure 10i), the peak value of the circulating currents is approximately 7.9 A, and it increases to approximately 8.5 A when the proposed control strategy is applied (see Figure 10j), representing an increase of 7.6%.

To demonstrate the reduction in the applied common-mode voltage depending on the control strategy, Figure 11 compares the desired values of the common-mode voltage during the conduced test of Figure 10. As mentioned before, these signals are generated directly by the DSP-based control system according to (25) and (27). As shown in Figure 11a, the maximum amplitude of the desired common-mode voltage is close to 120 V, while the

one generated with the proposed control strategy is close to 50 V, showing that this voltage can be reduced as the DC port is also reduced during LFM. Notice that during HFM, the desired common-mode voltage is similar in both cases because it corresponds to a third harmonic of the machine voltages, as is typical in machine drives.

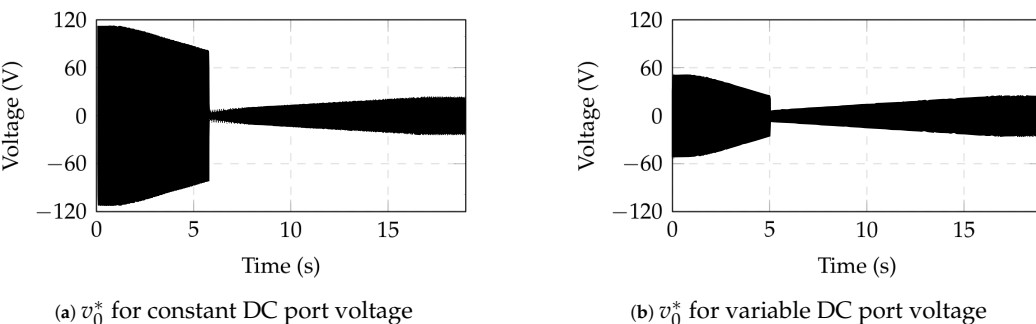

(a) $v_0^*$ for constant DC port voltage

(b) $v_0^*$ for variable DC port voltage

**Figure 11.** Comparison of the desired common-mode voltage for (**a**) constant DC port voltage and (**b**) variable DC port voltage.

## 6. Conclusions

In this work, the influence of the DC port voltage $E$ on the MMC behavior has been analyzed for both low- and high-frequency modes. During LFM, it has been demonstrated that a low value of $E$ is advantageous for the operation of the machine drive. For example, it could lead to a reduction in the voltage fluctuation in the MMC capacitors or the cell capacitance. Additionally, the common-mode voltage is reduced in the same proportion as $E$, without affecting severely the peak value of the circulating currents. The proposed state-space model of the dynamics of the cluster voltages as a function of the DC port voltage demonstrates that undesired dynamic changes can appear for sudden changes in $E$. Consequently, a closed-loop control system has been proposed to regulate the cluster voltages properly. In all the tests and comparisons with the conventional control strategies, the simulated as well as the experimental results obtained with an induction machine and a BTB-MMC show the effectiveness and feasibility of the proposed control schemes since they allow us to demonstrate that (1) the machine performance is the same for the proposed and conventional control strategies, (2) the fluctuations in the cluster voltages remain within the desired margin in the whole frequency range, (3) the injected common-mode voltage in the machine is reduced during LFM, and (4) there is not a significant increase in the circulating current when the proposed control strategy is applied.

**Author Contributions:** Conceptualization, M.E.; Methodology, R.C.; Software, M.E., E.E., F.D. and A.L.; Validation, M.E. and A.L.; Formal analysis, A.M. and R.C.; Investigation, M.E.; Writing—original draft, M.E.; Writing—review & editing, M.D. and E.E.; Supervision, R.C.; Project administration, R.C.; Funding acquisition, R.C. All authors have read and agreed to the published version of the manuscript.

**Funding:** The authors would like to express their sincere gratitude to the funding agencies that supported this work. First, support from Fondecyt Projects (Grant number 1221392 and 1231030) and Basal Project FB0008 Advanced Center for Electrical and Electronic Engineering were crucial in enabling the experiments and analysis presented in this paper. Additionally, the University of Costa Rica's group project C1467 contributed significantly to this project.

**Conflicts of Interest:** The authors declare no conflict of interest.

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
