# Peer review of "The Closed-Loop Control of the Half-Bridge-Based MMC Drive with Variable DC-Link Voltage"

_electronics, doi:10.3390/electronics12132791_

Round 1
Reviewer 1 Report
1. How does the proposed closed-loop control approach ensure a constant voltage fluctuation in the capacitors of the MMC during the whole operating range?
2. Can you explain in more detail the small-signal model that was developed to analyze the dynamic of the MMC when the magnitude of the dc-port voltage E is modified?
3. What are the main advantages and disadvantages of using a low value of E during LFM, as demonstrated in the paper?
4. How does the proposed closed-loop control system modify the dc-port voltage considering the dynamics of the capacitor voltages and not only steady-state expressions?
5. Can you explain how the proposed approach reduces the common-mode voltage during LFM and how this reduces the damage to the machine insulation or the leakage currents in the bearing?
6. What kind of experimental validation tests were conducted using the 7.5 kW experimental prototype composed of an induction machine fed by an MMC with 18 half-bridge cells?
7. How does the proposed closed-loop control system compare to conventional control strategies in terms of its effectiveness and feasibility?
8. Can you explain the behavior of the oscillating component of the cluster voltage v˜XcX (with X=P or N and x=a, b or c) during low-frequency modes?
9. How does the peak value of the circulating currents change when using the margin-based control strategy compared to the full-mitigation strategy?
10. What are the limitations of the machine-side MMC in terms of dynamic regulation and how does the proposed approach address them?
Minor Corrections are required.
Reviewer 2 Report
The aim of this work was the theoretical and experimental investigation of the influence of the dc-port voltage on the Modular Multilevel Converter under low and high frequency modes. Authors demonstrated that undesired dynamic changes can appear for sudden changes in E according to the state-space model of the dynamics of the cluster voltages. In order to regulate the cluster voltages authors used the closed-loop as an effective and feasible control system. The topic is original in the field. The effectiveness of the proposed solution was demonstrated according to simulation studies performed using a 7.5 kW experimental prototype based on an induction machine fed by an Modular Multilevel Converter with 18 half-bridge cells. The conclusions are consistent with the evidence and arguments presented. The references have been adequately cited and discussed. The paper is well written and is suitable for this Journal. Manuscript can be accepted for publication without any further changes.
Author Response
Dear reviewer, Thank you very much for your review.We are pleased that you considered our work good enough to be published.
Reviewer 3 Report
The article proposes a control for CMM with variable voltage on the DC link. Theoretical analysis is satisfactory, but more experimental results are needed. So my comments are:
1. Add the name of the variable to which it corresponds to the waveforms in Fig. 8.
2. Label figure 8 as (a) and (b) not left and right.
3. Add the waveform referring to the experimental common mode voltage using the proposed method and the conventional method
4. Is Figure 10 experimental? It looks like a simulation result. Add experimental results.
5. Add other DC link voltage variations and show the experimental results obtained.
Author Response
Dear reviewer,
We sincerely appreciate you taking the time to read our work. We have carefully considered your comments, and we are now providing our responses to address them. Please see the attachment.

Round 2
Reviewer 1 Report
Paper can be accepted in the present form.
No corrections.
Reviewer 3 Report
Thanks for the clarification, no more questions. Congratulations for the paper.